# Altered transcriptional and chromatin responses to rhinovirus in bronchial epithelial cells from adults with asthma

Britney A. Helling [1✉], Débora R. Sobreira[1], Grace T. Hansen [1], Noboru J. Sakabe[1], Kaixuan Luo[1], Christine Billstrand[1], Bharathi Laxman[2], Raluca I. Nicolae[1], Dan L. Nicolae[3], Yury A. Bochkov[4], James E. Gern [4], Marcelo A. Nobrega[1], Steven R. White[2] & Carole Ober[1]

There is a life-long relationship between rhinovirus (RV) infection and the development and clinical manifestations of asthma. In this study we demonstrate that cultured primary bronchial epithelial cells from adults with asthma (n = 9) show different transcriptional and chromatin responses to RV infection compared to those without asthma (n = 9). Both the number and magnitude of transcriptional and chromatin responses to RV were muted in cells from asthma cases compared to controls. Pathway analysis of the transcriptionally responsive genes revealed enrichments of apoptotic pathways in controls but inflammatory pathways in asthma cases. Using promoter capture Hi-C we tethered regions of RV-responsive chromatin to RV-responsive genes and showed enrichment of these regions and genes at asthma GWAS loci. Taken together, our studies indicate a delayed or prolonged inflammatory state in cells from asthma cases and highlight genes that may contribute to genetic risk for asthma.

[1] Department of Human Genetics, University of Chicago, Chicago, IL 60637, USA. [2] Section of Pulmonary and Critical Care Medicine, Department of Medicine, University of Chicago, Chicago, IL 60637, USA. [3] Department of Statistics, University of Chicago, Chicago, IL 60637, USA. [4] Department of Pediatrics, University of Wisconsin, School of Medicine and Public Health, Madison, WI 53706, USA. ✉email: bhelling@uchicago.edu

Human rhinovirus (RV) is a nonenveloped, single-stranded RNA virus that is responsible for more than half of occurrences of the "common-cold"[1]. Although RV often causes minor upper respiratory tract illnesses, RV infections can be more frequent and severe in people with respiratory diseases, such as asthma, chronic obstructive pulmonary disease, cystic fibrosis, and idiopathic pulmonary fibrosis[2–7]. In particular, RV infection in individuals with asthma is associated with greater lower airway inflammation and decreased lung function compared to individuals without asthma[2,3,8]. Moreover, RV-associated wheezing illness in the first 3 years of life is associated with increased risk of asthma later in childhood[9–11], suggesting that impaired response to RV may be an early manifestation of asthma in many children. The durable relationship between RV infection and asthma may be the consequence of shared genetic and/or environmental risk factors or of defective remodeling of the airway epithelium in response to RV that predisposes to the subsequent development of asthma[12].

Airway epithelial cells are the primary site of RV infection and replication, and release cytokines that orchestrate downstream responses[13–16]. It has previously been shown that bronchial epithelial cells (BECs) from asthma cases and non-asthma controls have different transcriptional and cytokine responses to RV in cell culture[17–21]. However, the mechanisms underlying these differences remain unclear. We hypothesized that modifications of the chromatin landscape in BECs from individuals with asthma are associated with altered transcriptional responses to RV. To address this hypothesis, we performed a multi-omic study of gene expression (by RNA-seq) and chromatin accessibility (by Assay for Transposase-Accessible Chromatin with high-throughput sequencing [ATAC-seq]) in the presence of either rhinovirus strain RV-A16 or vehicle in cultured human BECs, and of chromatin interactions (by promoter capture [pc]Hi-C) in ex vivo human BECs. Our study revealed profound differences in transcriptional and epigenetic responses to RV between cells from individuals with and without asthma. Using pcHi-C, we were able to predict the target genes of RV-responsive chromatin changes and identify correlated transcriptional and chromatin responses. These RV-responsive genes and correlated RV-responsive chromatin-gene pairs were also enriched for genes within asthma genome-wide association study (GWAS) loci[22]. Our study provides a first-time view of the relationship between transcriptomics, chromatin accessibility, GWAS and three-dimensional chromatin landscape in BECs exposed to an important viral trigger of asthma and asthma exacerbations. These data suggest a prolonged inflammatory response to RV in asthma cases and revealed potential gene targets and pathways involved in the aberrant BEC responses to RV in asthma.

## Table 1 Characteristics of bronchial epithelial cell donors.

|  | ATAC-seq and RNA-seq | | pcHi-C |
| --- | --- | --- | --- |
|  | Asthma cases (n = 9) | Controls (n = 9) | Pooled (n = 4 cases, 4 controls) |
| Sex |  |  |  |
| Female (%) | 5 (56%) | 4 (44%) | 2 (25%) |
| Male (%) | 4 (44%) | 5 (56%) | 6 (75%) |
| Age (years) |  |  |  |
| Range (mean) | 23–76 (46) | 48–82 (64) | 22–62 (44) |
| Ethnicity |  |  |  |
| Caucasian | 9 (100%) | 9 (100%) | 5 (62.5%) |
| Asian |  |  | 1 (12.5%) |
| Hispanic |  |  | 2 (25%) |

## Results

We characterized the transcriptional and chromatin landscape in human primary BECs from 18 adult lung donors, nine with and nine without asthma. Basal BECs were grown in submerged monolayer culture, triplicates of each were treated with either rhinovirus (RV) type A16 or vehicle control; cells were harvested for RNA-seq and ATAC-seq studies. Ex vivo BECs from eight additional donors (four asthma cases and four controls) were collected and processed for pcHi-C. Donor characteristics are summarized in Table 1 (individual data for the donors are included in Supplementary Data 1).

**Transcriptional responses to RV differ between asthma cases and controls**. To investigate BEC responses to a virus associated with worsening asthma and exacerbations, we assessed transcriptional responses to RV in cases and in controls. These analyses revealed significantly more RV-responsive genes in the controls (4645 genes; 2351 upregulated and 2294 downregulated; FDR < 0.05; Fig. 1A, Supplementary Data 2) compared to the cases (3381 genes; 1709 upregulated, 1672 downregulated; FDR < 0.05; Fig. 1B, Supplementary Data 2) (Fisher's exact test, $p < 10^{-5}$). Surprisingly, only 1929 genes showed significant responses to RV in both cases and controls at an FDR < 0.05 (~32% of all RV-responsive genes). Moreover, among the 1929 genes that were RV-responsive in both groups, the controls had overall larger responses (paired Wilcoxon sign rank test $p < 10^{-5}$), with median absolute fold-change among the 1929 genes of 1.5 in controls and 1.4 in cases (Mood's median test, $p < 10^{-5}$) (Fig. 1C). Among the 1929 shared RV-responsive genes, 13 responded in opposite directions in cases and controls (Supplementary Table 1), with the expression of 12 genes increasing in controls and decreasing in cases (ANKRD18B, C1orf131, CWC27, FAM72B, LEO1, MPHOSPH10, PSMC1P1, PTP4A2, RPIA, SET, SREK1, ZNF326) and one gene increasing in cases and decreasing in controls (ZMIZ1) in response to RV. Examples of different patterns of RV response in cases and controls are presented in Supplementary Fig. 1.

**Gene expression pathways that respond to RV differ between asthma cases and controls**. We next tested for enrichment of gene expression pathways and gene-ontology (GO) terms among the RV-responsive genes using iPathwayGuide (Advaita), considering pathways with Bonferroni adjusted $p$ values ≤0.05 as significantly enriched[23]. Four pathways were enriched in both the cases and controls (cytokine-cytokine receptor interactions, Epstein-Barr virus infection, Influenza A, and NOD-like receptor signaling pathway), six additional pathways were enriched in the controls but not in the cases (TNF signaling pathway, JAK-STAT signaling pathway, transcriptional misregulation in cancer, measles, herpes simplex virus 1 infection, and NF-kappa B signaling pathway) and one additional pathway was enriched in the cases but not in the controls (necroptosis) (Fig. 2A–B). Using GO terms, we identified 28 enriched biological processes that were associated with RV-responsive genes in both cases and controls (Supplementary Data 3). These shared GO terms included many processes centered on viral assault (e.g., response to external stimulus, response to other organisms, response to virus, defense response to virus, innate immune response, and immune response). RV-response genes in the controls were further enriched for 16 biological processes, many of which were related to cell death (Fig. 2C). In contrast, the 17 biological processes enriched for RV-responsive genes only in the cases, reflected viral replication and inflammation processes (Fig. 2D).

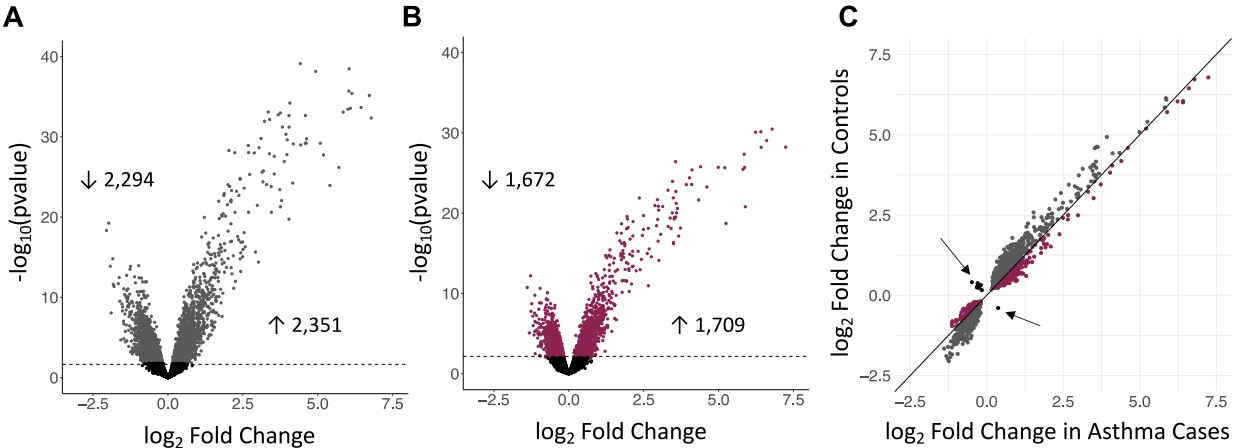

**Fig. 1 Different transcriptional responses to RV in asthma cases and controls.** Volcano plots showing the log-fold transcriptional changes to RV (*x*-axis) in controls (**A**) and cases (**B**) at FDR < 0.05 (dashed horizontal line). **C** The log fold change in the 1929 genes that are differentially expressed in both cases (*x*-axis) and controls (*y*-axis). Gray points indicate genes with greater log fold change in controls; burgundy points indicate genes with greater log fold change in cases. Arrows point to black dots representing the thirteen genes with opposite direction of effect in cases and controls.

**Chromatin responses to RV differ between asthma cases and controls.** Based on the large proportion of genes that responded to RV only in the cases or only in the controls, we expected to observe differences in chromatin responses to RV between cases and controls. The larger number of genes that responded to RV in the controls, and their enrichment in more diverse pathways and different biological processes, further suggested that chromatin changes would be more dynamic in the controls. Indeed, we identified 2458 RV-responsive regions of open chromatin in cells from the controls, but only 238 RV-responsive regions of open chromatin in cells from the cases at an FDR < 0.05 (Fisher exact, $p < 10^{-5}$; Fig. 3A–B, Supplementary Data 4). Despite our expectations, the actual number of differences was surprising (27% fewer transcriptional responses vs. 90% fewer chromatin responses in the cases compared to controls). Furthermore, only 58 regions were RV-responsive in both cases and controls, and those regions showed overall greater responses in the controls compared to the cases (Fig. 3C) (paired Wilcoxon sign rank $p = 0.013$). The striking reduction of chromatin response to RV in BECs from asthma cases compared to controls suggests either a loss of response in cells from asthma cases or a fixed state in the cases in which the chromatin is primed to be RV-responsive.

Next, we considered the sequences within the areas of RV-responsive chromatin and tested for enrichment of transcription factor binding sites using HOMER motif analysis[24], to identify the transcription factors binding sites (TFBS) that overlap with changes in chromatin accessibility. Regions with increased accessibility ($n = 153$ in cases and 1757 in controls) and decreased accessibility ($n = 85$ in cases and 701 in controls) were assessed separately to allow predictions of directionality. Fig. 4A shows the number of transcription factors with enriched binding motifs in areas of chromatin with increased or decreased accessibility in response to RV (Table of TFBS enrichment in Supplementary Data 5). The five most enriched TFBS are plotted against the background prevalence of the corresponding binding sites in areas of open chromatin in Fig. 4B–D. Additionally, 21–44% of predicted transcription factors are also RV-responsive in BECs. Overall, these results provide independent support of the ATAC-seq findings by showing that areas of open chromatin are enriched for transcription factors that are also RV-responsive.

We next explored the direction of effect for each RV-responsive transcription factors with enriched binding motifs. In-line with expectations, the majority of the transcription factors in areas with increased accessibility (more open chromatin) after

RV exposure in either cases (2 of 3; *IRF1* and *SP1*) or controls (6 of 7; *IRF1, IRF2, PRDM1, MYBL2, IRF3* and *CTCF*) also showed increased expression after treatment with RV. This is consistent with the prediction that genes regulated by these transcription factors would have enhanced expression after exposure to RV. In contrast, in areas with decreased accessibility (more closed chromatin) in the controls, only 10 of 20 corresponding transcription factors show decreased expression (*FOXP2, TFCP2L1, TP73, BCL6, HIF1A, FOXA3, FOXO1, STAT4, TP53*, and *EPAS1*). This implies that for half of the transcription factors at regions with decreased accessibility, there is both decreased accessibility of binding sites and decreased availability of transcription factors; for the other half there is increased availability of the corresponding transcription factor despite decreased accessibility of binding sites. Many of the transcription factors with enriched binding motifs at regions with RV-responsive chromatin have been implicated in studies of lung function (FOXP2[25], EPAS1[26], SP1[27]), immune function (IRF1[28], IRF2[29], IRF3[30], BCL6[31], PRDM1[32], FOXA3[33], FOXO1[34]), and asthma (CTCF[35], HIF1A[36], STAT4[37]).

**Chromatin–chromatin interactions link regions of RV-induced chromatin responses to their target genes.** Although epigenetic marks are often assigned to the nearest gene to assess functionality, it has recently been demonstrated that the majority of chromatin–chromatin interactions skip at least one gene[38]. To link areas of open chromatin in BECs to the promoters of their putative target genes, we performed pcHi-C[39,40] in freshly isolated cells from eight donor lungs that were not included in the cell culture studies (see Methods). Overall, we detected 601,845 chromatin-promoter interactions (Supplementary Data 6). Of these, over a third (211,080) overlapped with areas of open chromatin defined by ATAC-seq in the cultured BECs from cases or controls (treated with vehicle or RV), reflecting areas that are active or poised for transcription in these cells.

To further characterize these regions, we defined the pcHi-C-predicted target genes for RV-responsive areas of open chromatin, and then tested for correlations between the relative 'openness' of the chromatin and the transcript levels of the predicted target gene. We first considered the 4545 RV-induced transcriptional responses and 2458 RV-induced chromatin responses in BECs from controls. This revealed 189 regions (7.7% of the tested regions) of RV-responsive chromatin that

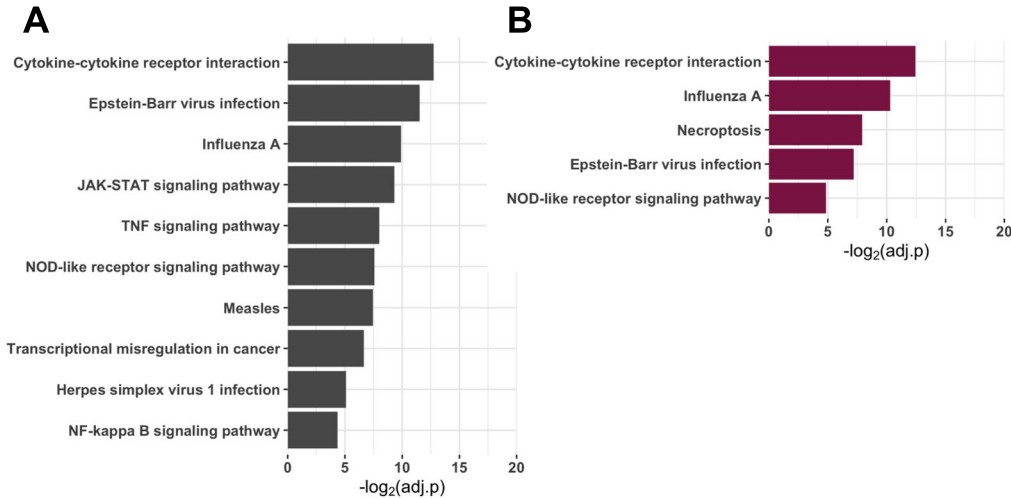

## C
### Biological Processes Uniquely Enriched in Controls

| GO ID | GO Term | Genes in GO | adj. p-value |
|---|---|---|---|
| GO:0042981 | regulation of apoptotic process | 442 / 1061 | 8.9E-05 |
| GO:0012501 | programmed cell death | 587 / 1460 | 1.6E-04 |
| GO:0043067 | regulation of programmed cell death | 444 / 1076 | 3.4E-04 |
| GO:0032103 | positive regulation of response to external stimulus | 100 / 190 | 5.5E-04 |
| GO:0010941 | regulation of cell death | 471 / 1157 | 9.5E-04 |
| GO:0008219 | cell death | 613 / 1549 | 1.1E-03 |
| GO:0006915 | apoptotic process | 548 / 1371 | 1.4E-03 |
| GO:0032879 | regulation of localization | 655 / 1678 | 4.2E-03 |
| GO:0048584 | positive regulation of response to stimulus | 601 / 1528 | 4.4E-03 |
| GO:0007165 | signal transduction | 1268 / 3437 | 1.1E-02 |
| GO:0010469 | regulation of signaling receptor activity | 119 / 247 | 1.6E-02 |
| GO:1902533 | positive regulation of intracellular signal transduction | 299 / 714 | 1.7E-02 |
| GO:0051239 | regulation of multicellular organismal process | 741 / 1935 | 1.9E-02 |
| GO:0023052 | signaling | 1352 / 3689 | 2.2E-02 |
| GO:0050663 | cytokine secretion | 72 / 136 | 3.1E-02 |
| GO:0007154 | cell communication | 1358 / 3712 | 3.2E-02 |

## D
### Biological Processes Uniquely Enriched in Cases

| GO ID | GO Term | Genes in GO | adj. p-value |
|---|---|---|---|
| GO:1903901 | negative regulation of viral life cycle | 41 / 68 | 6.5E-06 |
| GO:0048525 | negative regulation of viral process | 45 / 79 | 1.3E-05 |
| GO:0006950 | response to stress | 762 / 2613 | 1.1E-04 |
| GO:0045071 | negative regulation of viral genome replication | 32 / 51 | 1.2E-04 |
| GO:0043901 | negative regulation of multi-organism process | 59 / 125 | 5.5E-04 |
| GO:0002376 | immune system process | 563 / 1893 | 1.1E-03 |
| GO:0071346 | cellular response to interferon-gamma | 55 / 119 | 3.6E-03 |
| GO:1903900 | regulation of viral life cycle | 56 / 122 | 3.8E-03 |
| GO:0070887 | cellular response to chemical stimulus | 615 / 2109 | 5.9E-03 |
| GO:0051092 | positive regulation of NF-kappaB transcription factor activity | 55 / 121 | 7.1E-03 |
| GO:0002790 | peptide secretion | 130 / 361 | 1.5E-02 |
| GO:0032612 | interleukin-1 production | 33 / 62 | 1.8E-02 |
| GO:0045069 | regulation of viral genome replication | 40 / 81 | 1.8E-02 |
| GO:0040012 | regulation of locomotion | 213 / 649 | 2.3E-02 |
| GO:0032611 | interleukin-1 beta production | 29 / 53 | 3.5E-02 |
| GO:0032652 | regulation of interleukin-1 production | 29 / 53 | 3.5E-02 |
| GO:0009306 | protein secretion | 123 / 344 | 4.0E-02 |

**Fig. 2 Pathway and gene-ontology network enrichments for RV-responsive genes in cases and controls.** RV-responsive pathways in controls (**A**) and cases (**B**). GO identified biological processes enriched in controls (**C**) or cases (**D**). Twenty-eight additional terms were enriched in both cases and controls (see Supplementary Data 3). Bonferroni adjusted *p* values shown.

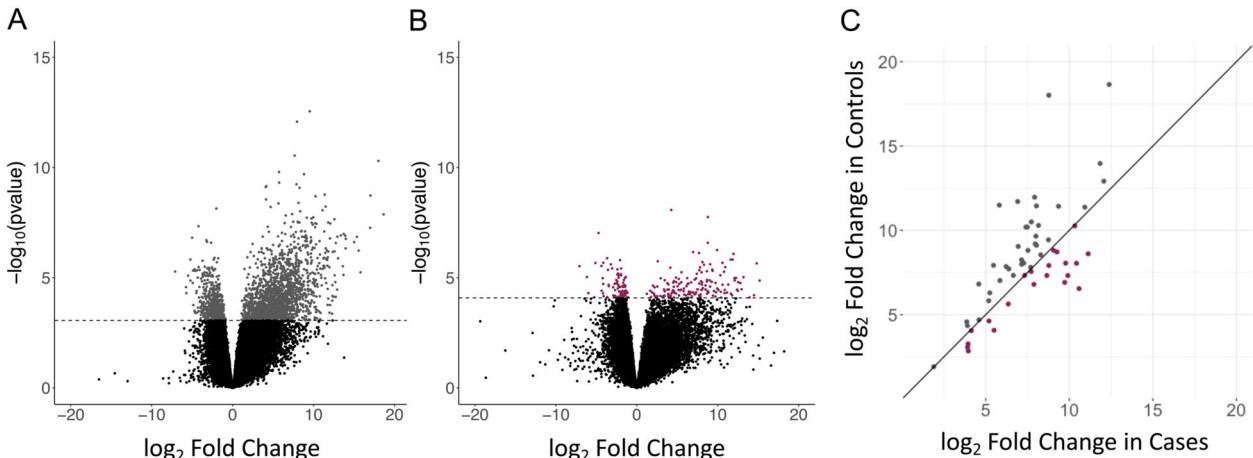

**Fig. 3 Chromatin accessibility response to RV.** RV infection results in 2458 differentially accessible chromatin regions in controls (five nonsignificant values were outside the range of the x-axis and are not shown) (**A**) and 238 in asthma cases (10 nonsignificant values were outside the range of the x-axis and are not shown) (**B**); (FDR < 0.05). **C** Fold changes for the 58 regions of RV-responsive chromatin accessibility. Gray dots represent greater change in chromatin accessibility in controls and burgundy dots represent greater change in cases.

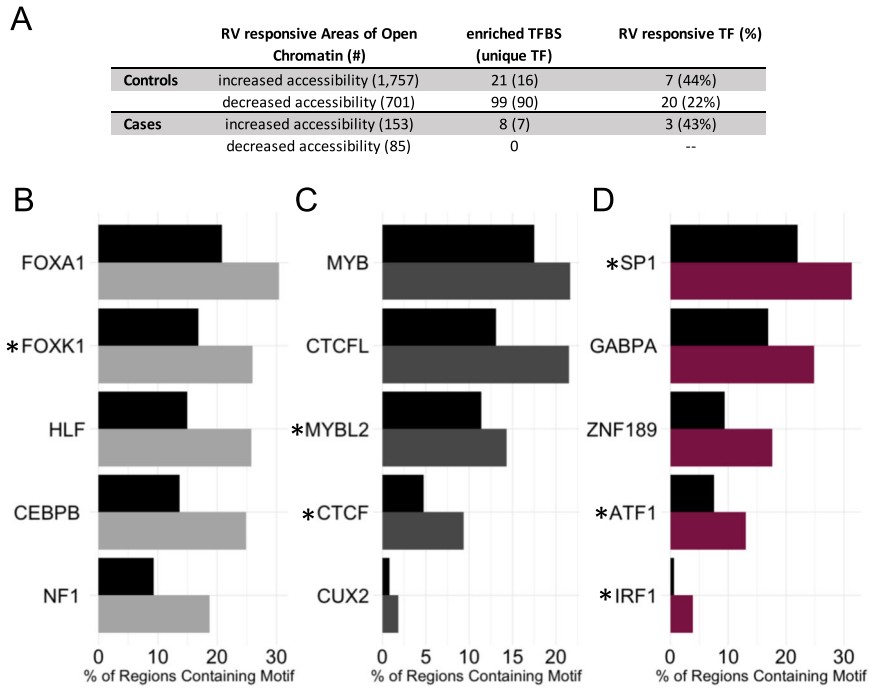

| | RV responsive Areas of Open Chromatin (#) | enriched TFBS (unique TF) | RV responsive TF (%) |
|---|---|---|---|
| Controls | increased accessibility (1,757) | 21 (16) | 7 (44%) |
| | decreased accessibility (701) | 99 (90) | 20 (22%) |
| Cases | increased accessibility (153) | 8 (7) | 3 (43%) |
| | decreased accessibility (85) | 0 | -- |

**Fig. 4 HOMER motif analysis shows enrichments of TFBS in areas of RV-responsive open chromatin. A** RV-responsive areas of open chromatin have been separated by those with increased accessibility and decreased accessibility in the RV-treated sample in the first column of the table. The number of enriched TFBS and the corresponding number of unique TF identified by HOMER (in parentheses) are shown in the second column of the table. The last column indicates the number (percent in parentheses) of unique transcription factors identified by HOMER that are also RV-responsive. **B** The five most enriched TFBS are shown for areas with increased accessibility in RV-treated samples from controls. The black bars indicate the % of regions containing the motif among all ATAC-seq peaks and the grey bar indicates the % of regions containing the motif in the RV-responsive areas of open chromatin. Panels **C** and **D** show the five most enriched TFBS for the areas with decreased accessibility in controls and increased accessibility in cases, respectively. *HOMER identified TFBS that correspond to transcription factors that also have RV-responsive gene expression.

were correlated with the expression of RV-responsive target genes (Spearman's correlation, adj. $p < 0.05$) (Fig. 5A, Supplementary Data 7). We next considered the 3381 RV-induced transcriptional responses and 238 RV-induced chromatin responses in BECs from the asthma cases. The relative 'openness' at 14 regions (5.9% of the tested regions) of RV-responsive chromatin were significantly correlated with the expression of RV-responsive target genes (Spearman's correlation, FDR adj. $p < 0.05$) (Fig. 5A, Supplementary Table 2). Of the 189 gene-chromatin pairs in

controls and 14 in cases, six were shared. An example of correlated RV-responsive chromatin accessibility and gene expression in controls is shown in Fig. 5B–C.

To determine whether chromatin is primed for response to RV in the cases, we next evaluated the 182 RV-responsive gene-chromatin pairs determined by pcHi-C in cells from controls that were not also paired in cells from cases because either the gene ($n = 1$), the chromatin ($n = 119$), or both ($n = 63$) were not RV-responsive in the cases. The 119 pairs in which genes were RV-

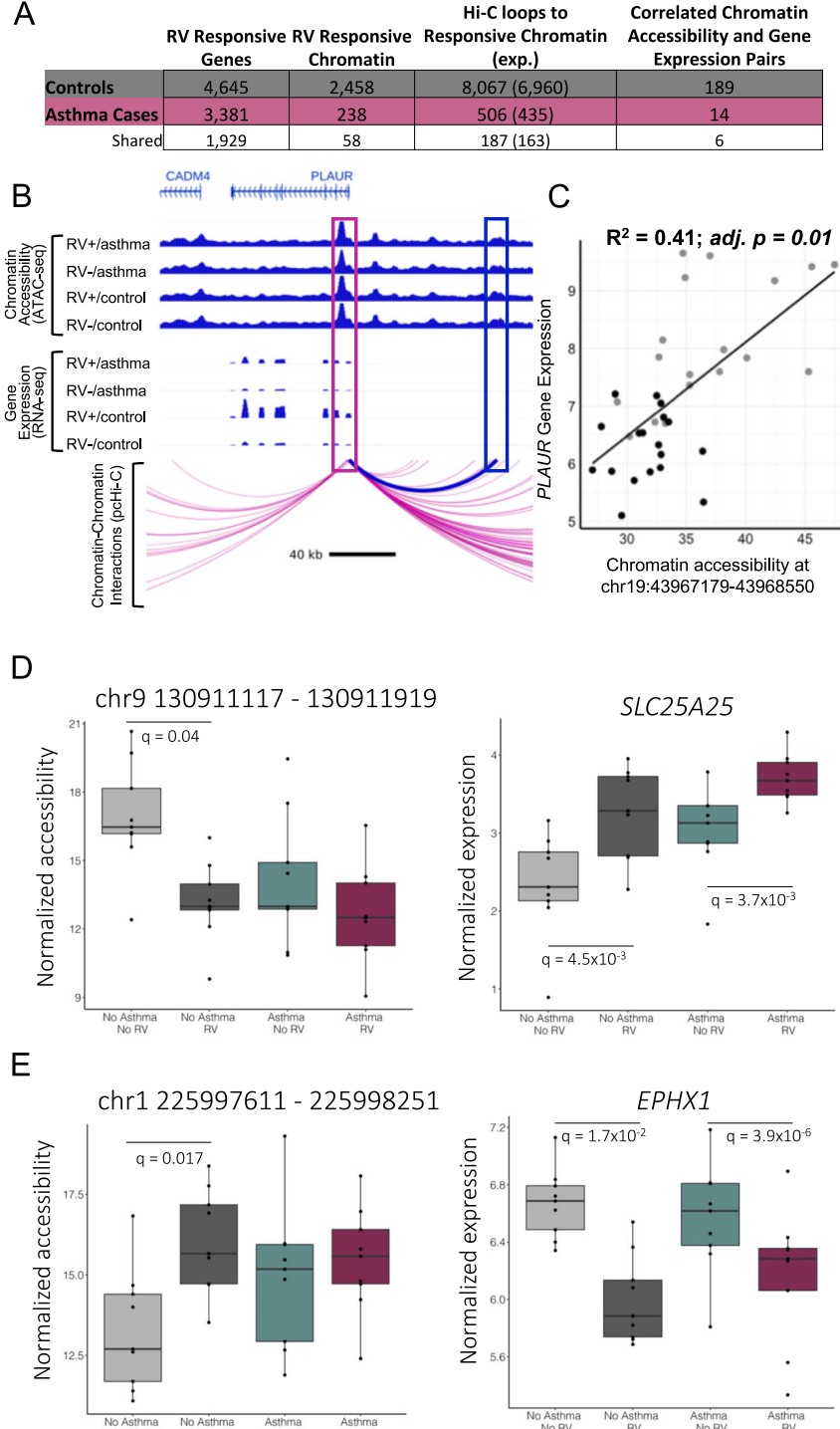

| | RV Responsive Genes | RV Responsive Chromatin | Hi-C loops to Responsive Chromatin (exp.) | Correlated Chromatin Accessibility and Gene Expression Pairs |
|---|---|---|---|---|
| **Controls** | 4,645 | 2,458 | 8,067 (6,960) | 189 |
| **Asthma Cases** | 3,381 | 238 | 506 (435) | 14 |
| Shared | 1,929 | 58 | 187 (163) | 6 |

**Fig. 5 RV-responsive areas of open chromatin correlated with gene expression at pcHi-C predicted target genes. A** The numbers of RV-responsive genes, chromatin regions, or pcHi-C target genes of RV-responsive chromatin regions in controls (gray) and cases (burgundy) and both (white) are shown in the first three columns. The last column shows the number the pcHi-C predicted genes where the chromatin accessible and gene expression were correlated (Spearman's correlation, FDR adj. $p < 0.05$). **B** The promoter of *PLAUR* loops to an area of open chromatin at chr19:43967179–43968550. Chromatin accessibility for each condition is shown in the top panel, gene expression in the second panel and the bottom panel indicates pcHi-C looping from the *PLAUR* promoter. The pink box indicates the *PLAUR* promoter, the blue box indicates the RV-responsive area of open chromatin. **C** Correlation between expression of *PLAUR* and chromatin accessibility at chr19:43967179–43968550. **D**, **E** are gene-chromatin pairs where the gene is RV-responsive in both cases and controls while the chromatin is only RV-responsive in the controls.

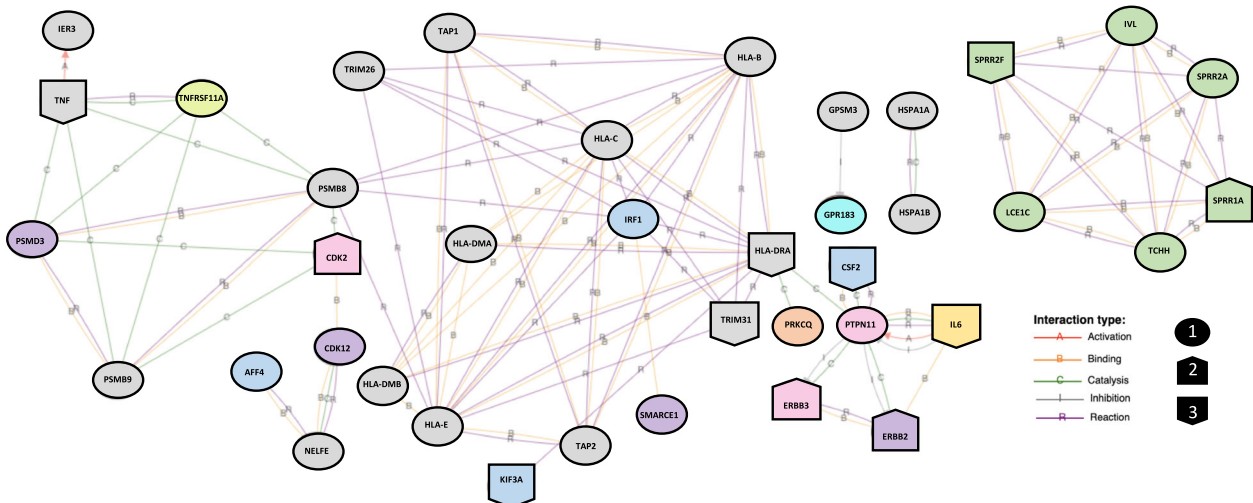

**Fig. 6 A total of 141 RV-responsive genes in cases or controls overlap with previously defined asthma-GWAS loci.** Thirty-nine of these genes interact with at least one other of the 141 genes, forming four networks annotated by iPathwayGuide. The genes are color coded to show genes within shared loci; gray genes are all within the HLA region on 6p21.32, purple are within 17q12, green are within 1q21.3, pink are within 12q13.2, yellow is within 7p15.3, orange is within 10p14, yellow–green is within 18q21.33, blue are within 5q31.1, and teal is within 13q32.3. The shape of the gene indicates differential expression between cases and controls (FDR < 0.05); shape 1 indicates no change in expression between cases and controls, shape 2 indicates increased expression in cases compared to controls, and shape 3 indicates decreased expression in cases compared to controls.

responsive in the cases but the ATAC-seq peaks that were connected by pcHi-C loops were not RV-responsive are examples of regions in which chromatin accessibility is responsive to RV only in the controls, despite their target genes being RV-responsive in both cases and controls, and potential examples of priming. Of the 119 areas of RV-responsive chromatin in the controls but not in the cases at q < 0.05, 75 showed some evidence of RV response in the same direction in cases and controls (the log fold changes corresponding to response were both positive or both negative in the cases and controls); 41 of the 75 chromatin responses were nominally significant at an unadjusted p values <0.05 in the cases. The remaining 44 areas of open chromatin had RV responses in the opposite direction in cases and controls, 24 of these opposite responses were also nominally significant in the cases. We consider the remaining 54 of the 119 areas of open chromatin without evidence of an RV response (unadjusted p > 0.05) that were connected to an RV-responsive gene as suggestive of chromatin that is primed (or poised) in the cases, allowing the gene to respond upon treatment. Thus, at least 45.4% of the 119 RV-responsive gene-chromatin pairs determined by pcHi-C in cells from controls that were not paired in cells from cases because of lack of chromatin response, show evidence of priming in the cases.

Figure 5D–E shows examples of priming for the *SLC25A25* and *EPHX1* genes, both of which have been previously associated with asthma phenotypes[41,42], and their Hi-C paired region of open chromatin (ATAC-seq peaks) in cases and controls. *SLC25A25* shows increased expression in response to RV in both cases and controls, while the interacting ATAC-seq peak (chr9 130911117–130911919) shows decreased accessibility after exposure to RV only in the controls. In both the RV and vehicle treated cells from the cases, chromatin accessibility is not responsive to RV and is similar to the RV-treated cells from controls, suggesting that the chromatin in cases is already primed. For the *EPHX1* gene, RV treatment results in decreased expression in both cases and controls, but the interacting ATAC-seq peak (chr1 225997611–225998251) is RV-responsive only in the controls. This area of open chromatin is more accessible in cells from controls with RV treatment compared to vehicle treatment and similar to both the RV and vehicle treated

cells from cases, again suggesting that the chromatin is primed for a response to RV in cells from the cases.

**Linking RV-responsive genes to asthma-GWAS loci**. To explore the possibility that RV-responsive genes may contribute to asthma genetic risk, we explored the overlap of the locations of RV-responsive genes and 61 GWAS loci for childhood onset or adult onset asthma[22]. GWAS loci were defined by LD using FUMA[43], as described in Pividori et al.[22]. Overall, 741 coding genes were annotated to the 61 loci, of which 294 genes were expressed in BECs in our study (Supplementary Data 8). Of the 4645 RV-responsive genes in controls and 3381 in cases, 116 (2.5%) and 83 (2.5%), respectively, were within asthma-GWAS loci. To test whether the overlap of RV-responsive genes and GWAS loci were more than expected by chance, we performed 100 permutations in R in which 4645 or 3381 genes were randomly selected from the 13,629 genes detected as expressed in BECs. The mean number of genes overlapping with GWAS loci was 101.2 (SD = 9.0) and 71.1 (SD = 7.0), respectively, compared to 116 overlaps observed in the RV-responsive genes in the controls and 83 overlaps observed in the asthma cases, suggesting modest enrichments of RV-responsive genes within GWAS loci (permuted p = 0.04 and 0.07, respectively).

Next, we asked whether the set of genes or chromatin regions participating in RV-responsive gene-chromatin pairs (Fig. 5A) were enriched within asthma-GWAS loci. Given the small number of pairs in the cases, we pooled the 14 pairs from cases with the 189 pairs from controls (197 unique pairs total, including 170 genes and 170 regions of open chromatin) and repeated the enrichment analysis described above, randomly selecting 170 genes from the 13,629 genes detected as expressed in the BECs. Among the 100 permutations, there was a mean of 3.74 (SD = 2.0) genes localized to asthma-GWAS loci, compared to six genes observed in our study (Supplementary Table 3), a suggestive enrichment (permuted p = 0.12). Finally, we repeated this for the ATAC-seq regions, randomly selecting 170 regions in each of 100 permutations. A mean of 1.57 (SD = 1.1) regions fell within GWAS loci in the permuted data set compared to eight observed regions in our study, a greater than 5-fold enrichment (permutation p < 0.01).

To further assess the potential role of these genes in functional networks, the 141 RV-responsive genes at GWAS loci were put into iPathwayGuide network analysis. Thirty-nine of the genes formed one large and three small interaction networks (Fig. 6). A large network of 29 genes, included the two of the RV-responsive gene-chromatin pairs (PSMB8 and PRKCQ) (Supplementary Table 3), four genes (out of 14 RV-responsive genes in this region) at the most statistically significant childhood onset asthma locus on 17q12 (ERBB2, PSMD3, CDK12, SMARCE1), four genes out of nine RV-response genes at two 5q31.1 loci (CSF2, IRF1, KIF3A, AFF4), and 15 HLA region genes (out of 39 RV-responsive genes in this region). Two smaller networks of two genes each included three additional HLA regions genes (HSPA1A and HSPA1B; GPSM3) in addition to one gene at the 13q32.3 locus (GPR183). The fourth network was comprised of six out of 25 RV-responsive genes at the Epidermal Differentiation Complex (EDC) on chromosome 1q21.3 (SPRR1A, SPRR2A, SPRR2F, IVL, LCE1C, TCHH), the second most statistically significant childhood onset asthma locus[22] (Fig. 6).

Lastly, we asked if any of these 39 RV-responsive genes within asthma-GWAS loci were also differentially expressed between cases and controls. In the vehicle treated samples, CDK2, CSF2, IL6, SPRR1A, SPRR2F, and TRIM31 were differentially expressed between cases and controls while in the RV-treated samples CSF2, ERBB2, ERBB3, HLA-DRA, IL6, KIF3A, SPRR1A, SPRR2F, TNF, and TRIM31 were differentially expressed between cases and controls (genome-wide FDR < 0.05) (Fig. 6, Supplementary Data 2). These data further suggest that RV-responsive genes are enriched for asthma-GWAS genes that also show transcriptional differences between cases and controls, suggesting that the differential response may be at least in part genetically determined.

## Discussion

Despite the important role that RV infection has in both asthma inception and progression; little is known about the underlying epigenetic mechanisms for interindividual responses to this common respiratory virus. To address this important question, we used a multi-omics approach to investigate differential transcriptional and epigenetic responses to RV in BECs from adults with and without asthma. Our study identified differences in epigenetic responses to RV in cells from asthma cases and controls, with significantly fewer responsive sites overall and smaller magnitudes of response at shared areas of RV-responsive chromatin in cells from asthma cases. Despite the dampened chromatin response, BECs from cases showed robust transcriptional responses to RV treatment, albeit less than in BECs from controls. This paradoxical observation suggests that the chromatin architecture in cells from asthma cases are in a fixed chromatin state that allows for robust transcriptional responses upon infection with RV. Whether this reflects airway epithelial cell damage due to more frequent RV-associated illnesses throughout life or whether this indicates an innate difference that brings about the life-long impaired responses to RV infection in individuals with asthma cannot be determined from this cross-sectional study. Nonetheless, the profound differences in transcriptional and chromatin responses to RV supports the hypothesis that epigenetic remodeling in the bronchial epithelium of asthma cases leads to impaired transcriptional responses to RV, and ultimately to more severe clinical outcomes in those with asthma.

The observed differences in gene expression response to RV in cases and controls were further highlighted by pathway and GO analyses. First, although there were four pathways which were enriched in response to RV in both cases and controls, six pathways were only enriched in the controls. Second, the enrichment of RV-responsive genes among apoptotic and cell death biological processes in the controls but of inflammatory and viral replication pathways in the cases was unexpected. In particular, these differences may reflect a delayed or prolonged response to RV in BECs from asthma cases. The natural course of RV infection in respiratory epithelium begins with an inflammatory phase followed by an apoptotic phase. Our data indicate that BEC responses to RV in controls are in the apoptotic phase whereas BEC responses to RV in cases are still in the inflammatory phase after a 24-h exposure. This suggests that BECs from asthma cases remain in a proinflammatory state for a longer duration than do cells from controls, or alternatively that these cells have a delayed activation of their response to RV. The latter possibility is supported by human in vivo studies of response to RV-A16 inoculation in subjects with allergic asthma after challenge with nasal allergens or placebo[44]. In those studies, subjects undergoing allergen challenges had similar infection rates but delayed cold symptoms after RV inoculation. This modified response may contribute both to the airway remodeling characteristics of the asthmatic airway as well as to sustained epigenetic changes in these cells.

Although innate immune memory was initially described in innate immune cells, other cell types, including epithelial stem cells, have been shown more recently to demonstrate responses consistent with innate immune memory[45,46]. The innate immune memory of BECs in response to RV could be due to greater frequency and prolonged inflammatory phase of RV infections, or even to other asthma-promoting exposures[47], such as other viruses, bacteria, allergens, air pollution, and cigarette smoke, among others. Innate immune memory due to epigenetic remodeling could contribute to the overall altered response of individuals with asthma to RV infections, and potentially account for the more severe clinical sequelae (i.e., hospitalizations, secondary infections, perceived symptom scores). Importantly, innate immune memory has been shown to be reversible in some cases[48], suggesting that an understanding of innate immune memory in BECs may lead to novel therapeutic targets for RV infections and its sequelae in individuals with asthma, or potentially to reducing the subsequent risk for asthma in young children.

The enrichment for RV-responsive genes within asthma-GWAS loci further suggests that some of the transcriptional differences observed between asthma cases and controls may be influenced by genetic differences that contribute to risk for asthma. The overlapping loci include RV-responsive genes at the two most significant childhood onset asthma loci on chromosomes 17q12–21 and 1q21.3[22], as well as the most significant shared locus between childhood and adult onset asthma at the HLA region, which accounts for 27.6% of all RV-responsive genes within asthma-GWAS loci. Other notable asthma-GWAS loci overlapping with RV-responsive genes are at chromosomes 2q12.1 (IL1R, IL1RL1, IL1RL2, IL18R1), 11q13.5 (LRRC32), 16p12.1 (IL4R), 16q12.1 (NOD2), and 18q21.32 (SERPINB2, SERPINB7, SERPINB10, TNFRSF11A) (complete list in Supplementary Table 3). Overall, these studies highlight genes at nearly 30% of asthma-GWAS loci that are RV-responsive in BECs, and potentially contributing to interindividual differences in asthma risk.

Our study significantly advances understanding the chromatin landscapes in BECs from asthma cases and controls and in response to RV. However, it has limitations. First, we looked only at a single timepoint post RV infection (24 h). It is possible that the observed differences in transcriptional and epigenetic responses to RV between cases and controls is a consequence of delayed, more rapid, or prolonged responses in the cases. That is, the response in cases at 24-hours may more closely resemble a

later or earlier post-infection timepoint response in controls. In fact, the enrichment in inflammatory GO terms in cases and enrichment for apoptotic GO terms in controls suggest a delayed or prolonged response to RV in asthma cases. Future time course studies of response to RV in asthma cases and controls can directly address this question. Second, our sample size of nine cases and nine controls was underpowered to look directly for genetic effects on RV response. Nonetheless, we see an enrichment of RV-responsive genes within asthma-GWAS loci, including those that were targets of RV-responsive chromatin accessibility sites. Third, although we use pcHi-C to wire RV-responsive ATAC-seq peaks to their RV-responsive gene targets, the pcHi-C was performed in cells from different individuals as those used in the cell culture studies and in ex vivo cells compared to cultured cells in this study. Differences between the donors with respect to sex ratio or other features, or differences in ex vivo vs. in vitro cells may have resulted in our underestimating the number of interactions present in the cultured and RV-treated cells. Lastly, our study focuses solely on the basal cell subtype of BECs. Because basal cells express the RV-A16 receptor (ICAM1), we chose to focus on this population of epithelial cells in this study. Future assessment of bronchial epithelial response to RV in a differentiated cell model will add further to our understanding of the effects of RV infection in other BEC types.

To our knowledge, this study provides the first description of pcHi-C in BECs, and of ATAC-seq in BECs from individuals with asthma. Utilizing these unique datasets, we were able to link potential functional changes in the epigenome to target genes and show correlations between chromatin accessibility and gene expression, and RV-responsive genes to asthma-GWAS loci. The data provided in this study are important not only to asthma but could also inform studies of other diseases of the lung epithelium. Finally, the results of our studies suggest that epigenetic priming and trained immunity may underlie the differential responses to RV infection in asthma cases and controls, and that these differences may reflect a prolonged inflammatory phase in BECs from individuals with asthma. Both priming and prolonged inflammation have previously been reported in asthma cases in response to RV-A16[49,50]. Together with the enrichment of asthma-GWAS loci among RV-responsive epigenetic sites, these results suggest potentially novel therapeutic targets that could ameliorate the downstream consequences of RV infection or possibly prevent the inception of asthma in at-risk individuals.

## Methods

**Sample collection and cell culture**. Lungs from 18 donors, not used for transplantation, were collected from the Gift of Hope, Organ and Tissue Donor Network, transferred to the University of Chicago, and processed within 48 h of harvest. BECs were isolated from the main bronchus as previously described[51]. Medical information, including sex, age, whether the donor had asthma, and cause of death were collected on all donors and group data are summarized in Table 1; individual donor data is shown in Supplementary Data 1.

BECs from the 18 donors were cultured on collagen coated flasks in F-media[52] until 80–90% confluency before being transferred to 12-well collagen coated transwell membranes. Cell culture experiments were conducted in basal cells, a subset of BECs that express ICAM1, the main receptor for RV-A. After reaching confluency (3–5 days), cells were treated with either $10.2 \times 10^6$ plaque forming units of RV-A16 in 5.1 ul of PBS or 5.1 ul of PBS (vehicle control). The RV was propagated and purified at the University of Wisconsin, Madison as previously described[53]. At harvest, cells were disassociated from the culture surface using Accutase (Sigma–Aldrich) and quantified using a hemocytometer and trypan blue to differentiate live and dead cells. All cultures had greater than 90% viability at the time of the harvest. After quantification, 5000 cells from each culture well were immediately processed for ATAC-seq and the remaining cells were resuspended in RLT buffer and stored at 4 °C for less than four hours prior to RNA and DNA extraction using the Qiagen AllPrep Mini Kit (Qiagen 80204).

**RNA-sequencing, QC, and analyses**. RNA quality and concentration were assessed using an Agilent 2100 bioanalyzer. RIN scores were 7.8 or better for all samples, with a mean RIN of 9.6. RNA was converted to cDNA using the SMART-

Seq v4 Ultra Low Input RNA kit for sequencing (Takara Bio cat. 634898). Standard library preparation was completed using the Illumina Nextera XT DNA Library Preparation Kit and library quality and concentration were assessed using an Agilent 2100 bioanalyzer. Indexed samples were pooled and sequenced on the Illumina Hi-Seq 4000 with 100 base-pair, paired-end sequencing to a minimum of 10 million mapped reads per sample and a mean of 14 million reads per sample. RNA-seq reads were mapped to genome assembly GRCh37(hg19) reference sequence using STAR (version 2.6.1)[54]. Read counts were adjusted to counts per million and normalized using TMM normalization[55]. PCA was conducted in R (version 3.6.2) using prcomp to select covariates to adjust for in the model; library prep technician, cDNA concentration, cDNA prep date, library concentration, and culture plate orientation were significant and adjusted for in the model. Voom was used to adjust for significant technical covariates[56]. Each sample was studied in triplicate (biological replicate), which was statistically accounted for using duplicateCorrelation in the Limma R package version 3.5 (https://bioconductor.org/packages/release/bioc/html/limma.html). Genes significantly associated with RV treatment or asthma were identified using Limma linear mixed modeling in R, while including sex as a covariate and blocking by sample to pair RV and vehicle treated samples from the same individual. An FDR-adjusted p value of 0.05 was used as a significance threshold, using the method of Benjamini and Hochberg[57].

**ATAC-sequencing, QC, and analyses**. For each sample, 5000 cells were processed immediately using the Nextera DNA Library Prep Kit (Illumina FC-121-1030) after collection using a standard ATAC-seq protocol[58] with the following modifications: the transposition reaction was incubated on a heated shaker at 37 °C for 30 min at 1000 rpm and the Qiagen MinElute PCR Purification kit was used to clean up the transposition reaction. Samples were indexed, pooled, and sequenced on the Illumina Hi-Seq 4000 with 50 bp single-end reads. Sample pools were resequenced until each sample had a minimum of 50 million mapped reads. Of the 108 samples sequenced, two samples from different donors failed QC. Two RV-treated samples from asthmatics were excluded (ID numbers: 7, 4); one was an outlier in several PCs including in PC1 and one did not have sufficient DNA concentrations for library prep (undetectable levels by bioanalyzer). Sequencing data that passed QC was mapped onto hg19 using bowtie2 version 2.3.2 (http://bowtie-bio.sourceforge.net/bowtie2/index.shtml). ATAC-seq peak calls were determined using MACS2 (version 2.1.0)[59]. Using bedtools (version 2.26.0), peak calls from all samples were pooled across all samples and overlapping or duplicate peaks were consolidated into a single peak that ranged from the 3′ most base to the 5′ most base in all overlapping peaks to create a peak-library or collection of all of the regions of accessible chromatin ($n = 142,120$ regions on autosomes) (raw and processed data available in GEO). Next, the number of reads aligned to each peak were determined for each sample and normalized to counts per million peak-library mapped reads. The fraction of reads in peaks or FRiP score was 7.2% across samples with 17.7% of peaks at Roadmap predicted transcriptional start sites[60] and 70% of peaks overlapping with ENCODE transcription factor binding sites[61]. PCA was conducted in R using prcomp to identify potential confounding of biological and technical covariates; number of additional PCR cycles and cell culture plate location were significant and adjusted for in the model. Each sample was part of a biological triplicate (except the two samples in which one replicate each did not meet QC criteria, discussed above). Replicate samples were adjusted in analyses using duplicateCorrelation and RV and Vehicle treated samples were paired using the blocking option as implemented in Limma. Peaks that were associated with asthma or RV infection in cases or controls were identified using Limma with a threshold of significance set at an FDR-adjusted $p$ value of 0.05, using the method of Benjamini and Hochberg[57].

**iPathwayGuide analyses**. Advaita iPathwayGuide (version v1906) was used to determine enriched pathways, and biological gene-ontology (GO) terms for both RV-responsive genes and asthma-associated genes in the vehicle treated and RV-treated samples (Supplementary Data 3)[23]. Multiple testing correction for pathways was addressed within the iPathwayGuide program by Bonferroni correction. To determine enrichment for pathways and GO terms in subset gene lists, specifically the RV-responsive genes within GWAS loci, were assessed through ToppGene instead of iPathwayGuide due to the input criteria needed for the latter.

**HOMER motif enrichment**. Using HOMER Motif Analysis (HOMER version 4.9.1), we identified transcription factor binding motifs that were enriched within the asthma-associated or RV-responsive DACs, using all of the areas of open chromatin as the background and a Benjamini–Hochberg FDR < 0.05[24]. Column descriptions for HOMER Motif Analysis in Supplementary Data 5 are included in Supplementary Table 4.

**Promoter capture Hi-C**. BECs were harvested from eight additional lungs not used for lung transplant through the Gift of Hope, Organ and Tissue Donor Network. The lungs were collected from four asthmatic and four non-asthmatic donors and processed within 24 h of harvest. The clinical data for these eight individuals are summarized in Table 1; individual level data are in Supplementary Data 1. BECs were processed for promoter capture [pc]Hi-C and analyzed as previously described[38]. In short, HiCUP version 0.5.9 was used to map pcHi-C reads to the

genome and remove technical artifacts. To obtain promoter interaction maps, CHiCAGO version 1.6.0 was performed on the pooled set of all reads from each sample and significance was defined as a CHiCAGO score >5, as recommended[62]. *NAA38* was removed from the analysis because of ambiguity on chromosomal location of the gene, which resulted in *NAA38* interactions being more distant than any other interactions.

**Testing for correlation between open chromatin and gene expression**. For these analyses, the mean was calculated for biological replicates for all protein coding genes and regions of open chromatin, and correlations between chromatin accessibility and gene expression was assessed by Spearman's rank-order correlation and using Benjamini–Hochberg FDR < 0.05 to adjust for multiple testing. Both cases and controls were used to assess correlation to increase sample size and for correlations that are observed in both cases and controls, the adjusted p value is slightly different based on number of tests conducted.

**Statistics and reproducibility**. Triplicates of each culture condition were conducted in each BEC line. The R package limma was used for all linear regression modeling for both RNA-seq and ATAC-seq analyses and an FDR-adjusted $p < 0.05$ was considered statistically significant. For iPathwayGuide GO term enrichment, enrichment was calculated by iPathwayGuide and a Bonferroni adjusted $p < 0.05$ was considered statistically significant. Permutation analyses were conducted in R for GWAS enrichment and a finding of fewer than five of 100 permutations reaching the observed number were considered statistically significant. To assess correlation between chromatin accessibility and gene expression in pcHi-C pairs, Spearman's rank-order correlations were conducted and Benhamini-Hochberg FDR < 0.05 was considered statistically significant. For HOMER motif enrichment, a Benjamini–Hochberg FDR < 0.05 was considered statistically significant.

**Reporting summary**. Further information on research design is available in the Nature Research Reporting Summary linked to this article.

## Data availability
RNA-seq, ATAC-seq, and pcHi-C data generated during this study are available in GEO (the Gene Expression Omnibus) under the reference series GSE152550. Results files for RNA-seq, ATAC-seq, pcHi-C, gene-chromatin correlations, GWAS assessment, gene-ontology analysis, and motif analyses are included in detail in Supplementary Data 1–8 and Supplementary Tables 1–3.

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

## Acknowledgements

We would like to thank the Gift of Hope, Organ and Tissue Donor Network and the generous donors for providing the lungs that makes this work possible. This work was supported in part by NIH grants U19 AI095230, R01 HL129735, and UG3 OD023282 and with resources provided by the University of Chicago Research Computing Center.

## Author contributions

Conceptualization, B.A.H and C.O.; Methodology, B.A.H.; Software, B.A.H., G.T.H., N.J. S., and K.L.; Formal analysis, B.A.H.; Investigation, B.A.H., D.R.S., C.B., and R.I.N.; Resources, Y.A.B., J.E.G., B.L., and S.R.W.; Writing—original draft, B.A.H. and C.O.; Writing—review and editing, B.A.H. and C.O.; Visualization, B.A.H.; Supervision, D.L. N., M.A.N., S.R.W., and C.O.; Project administration, B.A.H. and C.O.; Funding acquisition, C.O.

## Competing interests

J.E.G. reports consulting fees from Regeneron, consulting fees and stock options from Meissa Vaccines Inc and consulting fees from MedImmune/AstraZeneca outside the submitted work. In addition, J.E.G. has two patents on methods for propagating rhinovirus-C species viruses. The remaining authors declare no competing interests.
