## [Peer Review File · Communications Biology]

Reviewers' Comments:

Reviewer #1:

Remarks to the Author:

This manuscript from Hellig et al explores the relationship between asthma and RV response in BEC lines and ex-vivo lung cells using RNA-seq, ATAC-seq, and promoter-capture HiC. Overall, this is a large set of data that is potentially of interest to understand how asthma contributes to regulatory element regulation and transcriptional response to RV insult, which should be useful to the community. In parts, the manuscript suffers from lack of clarity and a coherent message tying together the large amount of data. At times, the logic was difficult to follow and I had to work very hard to understand some of the data presentation (particularly Figure 5). I have several comments below that may help improve the analysis in this submission, and I hope the authors will find these comments useful.

Line 49 – the authors are not directly testing epigenetic modification with the techniques used in this study. It would be more appropriate to make hypotheses about chromatin accessibility when using the tools herein.

Line 83 on (RNA-seq analysis) – From the text, it seems that the authors use only statistical significance and not a fold-change cut off to determine differentially expressed genes across RV or control treatment of BEC. I have a few questions about this. First, most of the RV-induced changes seem quite modest in both BEC from control and asthma patients. Can the authors explain why they do not consider a fold-change cut off, which may provide greater insight into pathway dysregulation? Second, the authors state RNA-seq in triplicate from 9 donor BEC lines, both RV treated and un-treated. I assume there must be some degree of variance in cell line response, given the large age range and other biological variables. What is the extent of this variance, and how does variance among samples compare with the relatively modest variance between control and asthma BEC response to RV? Associated, the authors should provide statistics for Figure S1.

Line 116 on (ATAC-seq analysis) – I have a few comments here.

1. The authors should show some of this ATAC-seq data. What are the quality scores (FRiP or TSS enrichment)? How well do data sets correlate across patient-derived BEC within the control and asthma groups?
2. The authors have used statistics to differentially call regions with altered accessibility. What if they took an RNA-centric approach here? For the genes that are upregulated in control with RV treatment, what happens to their promoters? For the asthma group? How do these regions compare between the two groups at the steady state without RV treatment? These kinds of analyses would allow the authors to address the question posed in line 128 – is there a lost response, or are the promoters in asthma patients “primed”?
3. The authors should provide details of their sequencing experiments – how many reads per sample, data normalization, correlation between data sets, etc. I may have missed this.
4. The figure legend for Figure 4 is incorrect and incomplete. The figure cannot be understood from the figure legend alone.
5. Also, here the authors have the opportunity to better integrate their data sets. They make the point that some of the TFs whose binding motifs are enriched from their HOMER analysis are dysregulated, but are they upregulated? Downregulated? How much? Is this a greater trend for the HOMER analysis integrated with RNA-seq, or just a handful of factors? Is the identity of these TF interesting from the perspective of lung function or related viral response?

Line 149 - Was there a reason for not using the same lines for the RNA/ATAC and pChIC?

Line 173 – I’m not a GWAS expert, but these results seem modest. Hopefully, the editors have recruited another reviewer that can better comment on this section of the manuscript.

Reviewer #2:

Remarks to the Author:

Summary

Helling *et al* present an interesting asthma genomics study using primary bronchial epithelial cells (BECs) stimulated by rhinovirus. They demonstrate that the response to rhinovirus is stunted in asthma BECs, and which could be mediated by a delayed or defective response. Similarly, ATAC-seq shows fewer changes in chromatin in response to rhinovirus infection. Some of the ATAC-seq regions correlate with the expressed genes. About a 3rd of detected promoter (pHi-C) – open chromatin interactions overlapped open chromatin detected in the BECs (with potentially poised transcription factors). There was at least modest enrichment of both rhinovirus responsive BEC genes and rhinovirus responsive chromatin among known asthma GWAS loci. Overall this study helps to dissect the mechanistic relationship between rhinovirus infection and asthma, and suggests that some of these responses could be genetically influenced by variants at GWAS loci.

Required changes

General

1. There is poor differentiation between types of fold-changes. In every case be clear whether you mean a natural log, log base 2, or log base 10. This is particularly important because the VOOM results in several places are reported as “logFC”, which leaves the reader guessing which is the correct scale.

Introduction

2. “... such as asthma, COPD, cystic fibrosis, ...” Abbreviation used without definition. Suggest changing to chronic obstructive pulmonary disease since it doesn't appear to be used again.

Results

3. “... majority of chromatin-chromatin interactions ship at least one gene...” You detected ~600k chromatin – promoter interactions. What percent of your data skip at least one gene? Not strictly necessary, just curious.

4. “... revealed 193 regions... that were correlated with the expression of RV-responsive target genes...” How did you define this? Maximum distance between gene start / end and ATAC peak? Overlap between ATAC and gene start end? Must be the closest gene to the ATAC peak?

5. “... we performed 100 permutations in which 4,645 or 3,381 genes were randomly selected from...” How? Python? R? Another tool?

Methods

6. “RNA- and ATAC-Sequencing, QC and analysis” Suggest splitting this into a sub-section on RNA-Seq and a sub-section on ATAC-Seq. Less confusing that way.

7. “...completed using the Illumina Nextera XT DNA Library Preparation Kit...” Obviously a DNA kit doesn't work for RNA. It would for cDNA. Did you use a kit or homemade method for cDNA generation? Was it ribosomal depletion or poly(A) mediated? Or did you use a different kit for RNA-Seq?

8. “...reference sequence using STAR.” Version of RNA-STAR missing.

9. “PCA was used to select covariates...” How? R? If so prcomp or princomp? Which R version? Same issue in the ATAC-Seq section.

10. “... the transposition reaction was incubated...” Commercial TN5, kit version, AddGene, or

other?

11. "...Qiagen MinElute PCR Purification kit was use to clean up transposition reaction." No bead size selection?

12. "ATAC-seq peak calls were determined using MACS2." No version.

13. "Peak calls from all samples were pooled across all samples and overlapping..." How? MACS2? Bedtools? R? Python? Version?

14. "Advaita iPathwayGuide was used to determine..." No release / version number.

15. "Significance was defined as a CHiCAGO score > 5." Is this their native negative log-weighted p-value, i.e. significance was P approximately 10^{-5} ?

Data availability

16. "... will be deposited in GEO." Publication should obviously not happen before the data are deposited in GEO or dbGAP. Sequencing data is identifiable and depending on the approvals surrounding these donations GEO might not allow their deposition.

Figures

17. Figure 1 – The x-axis is just log fold change. What log scale? The y-axis is $-\log_2$ p-value. This is a non-standard volcano plot scale, which usually uses $-\log_{10}$ adjusted P. I believe this is a typo. $-\log_2(0.05)$ is 4.32. $-\log_{10}(0.05)$ is 1.30. The horizontal line sure *looks* closer to 1 than 4. "The absolute log fold change..." there are negative values, so it can't be absolute fold-change.

18. Figure 2 – Again shows a $-\log_2$ (adj. P) scale. Is it log2 or log10? In sub-panel D, I understand the consistent color usage to help tie figures together. Personally I find the alternating colors hard to read in this plot. Purely personal preference, but you should consider not highlighting or just using an alternating light gray.

19. Figure 3 – Another unspecified log fold-change.

20. Figure 4 – This is overall confusing. I try to reach the metric that each figure should be interpretable without the accompanying text in the main document. The legend of 4 doesn't help much. The sub-panel letters don't match the figure. You need to tell the reader what the different bars mean (black is genomic background frequency?) in B-D. In table 4a there are some spacing issues in the "enriched TFBS (unique TF)" column where there is no space between the number and the parenthetical. You might want to change the last sentence to something like "Asterisks indicate transcription factors that have significant differential expression between cases and controls..."

Suggestions

Introduction

21. "... in the presence of RV-A16 or vehicle..." Suggest something like "... in the presence of rhinovirus strain RV-A16 or vehicle..." for clarity.

Methods

22. "... (n=142,120 regions on autosomes)(raw data available upon request)." This is the kind of non-identifiable data that would work well as a FigShare or other data sharing service deposit. Then you don't have to keep up with the data and can easily point people to the peak info with counts.

Tables

23. Many of the supplemental table fields I only understand because of familiarity with these types of analysis. Suggest adding a supplemental document that for each table defines each field to remove any ambiguity for non-experts.

24. May be easiest to combine *all* supplementary tables into a single workbook with multiple sheets, rather than dividing among CSV, workbook, and PDF.

25. Supplementary Table 3 – as an aside, we similarly sometimes see discordant fold-changes. I agree that for the sake of transparency it's worth listing them. Quick examination of the p-values show adjusted Ps quite close to the cutoff. Probably false-positives.

26. Supplementary Table 4 – Suggest reducing the table to only ontologies significant in one or the other. I tend to keep full gene lists for differentially expressed genes. But the large number of p-value == 1.0 make it difficult to sift through this table.

27. Supplementary Tables 9 and 10 – might be helpful to include the annotated gene start and end so we know how close these peaks are (or if they're overlapping with) the gene.

Discussion

28. This is mostly an aside. The inflammatory response is prolonged in asthma BECs, but it's also lower than in controls. Could the poor response lead to lack of ability to clear the RV? Leading to prolonged, though lower peak expression, of inflammatory genes causing remodeling?

Reviewer #3:

Remarks to the Author:

This study is conducted upon basal cells isolated from subject lungs, could the authors describe whether there were compositional differences between the cells obtained from either individual donors or between subject groups?

Cells from different donors were used for the HiC experiments, these donors had a different ratio of gender and ethnicity. Could the authors please comment in the implications of this for data interpretation.

There was quite an age range in the donors, did the authors observe any epigenetic differences between cells from young and old and how might any differences these contribute to interpretation of data, could the authors describe any medication that the donors were on prior to death.

Throughout the manuscript it is not clear what inherent differences there are between the study populations prior to rhinoviral infection. If there are fundamental and consistent differences in epigenetic landscapes between the subject groups then it follows that transcriptomes and ATAC profiles will be different following infection. Thus, could the authors describe which pathways were differentially expressed at baseline in order to separate asthma from infection.

In figure 4 data shows enriched TF motifs in ATAC regions, in response to RV. Were any of these motifs found in combination? What percentage of regions lacked any of the motifs shown?

In my opinion the Hi C data is under-reported and under-interpreted in this study, I'd have preferred to see some overall interpretations of the data and would imagine that the scale of the data is beyond what can be reported as a part of a manuscript, I would therefore assume that this might be followed up in future publications. First for instance, description of the relatedness of data from each of the donors within each group and fundamental differences between subject groups. The authors might describe differences between opening, closing and stable interactions in response to RV etc before arriving at the data presented

Reviewers' Comments	Authors' Responses
We thank the reviewers for their careful review of our manuscript and their very comments and suggestions. We have responded to each comment below and made changes in the manuscript when indicated (see red text). We think our manuscript is significantly improved based on these suggestions. We also note that we changed Figure 5 to provide more visual examples of the relationship between gene expression, chromatin accessibility, and pcHi-C. This updated figure has been included in the resubmission as well as at the end of this response.	
Reviewer 1	
Line 49 – the authors are not directly testing epigenetic modification with the techniques used in this study. It would be more appropriate to make hypotheses about chromatin accessibility when using the tools herein.	The reviewer makes a good point. We have deleted “epigenetic” and now state the hypothesis as follows (line 46): “We hypothesized that modifications of the chromatin landscape in BECs from individuals with asthma are associated with altered transcriptional responses to RV.”
Line 83 on (RNA-seq analysis) – From the text, it seems that the authors use only statistical significance and not a fold-change cut off to determine differentially expressed genes across RV or control treatment of BEC. I have a few questions about this. First, most of the RV-induced changes seem quite modest in both BEC from control and asthma patients. Can the authors explain why they do not consider a fold-change cut off, which may provide greater insight into pathway dysregulation?	This is correct, we used only an FDR cut-off. The reason for this was two-fold. First, using a fold-change cut-off will bias significant findings to more lowly expressed genes, for which smaller changes in gene expression result in larger fold-changes. Second, we wanted to be more inclusive because: (i) we don't know how much gene expression change is biologically significant, and (ii) we wanted to cast a wider net and include more genes because the downstream uses of the differentially expressed genes were for pathway analysis and for examining correlations between differentially expressed genes and chromatin accessibility.
Second, the authors state RNA-seq in triplicate from 9 donor BEC lines, both RV treated and un-treated. I assume there must be some degree of variance in cell line response, given the large age range and other biological variables. What is the extent of this variance, and how does variance among samples compare with the relatively modest variance between control and asthma BEC response to RV?	We appreciate this question. There are indeed donor effects, which we accounted for in our models as described in the following. We also show below results of the principle component analyses (PCAs) after adjusting for technical covariates —as noted in the methods (lines 378-381 and 411-417)— to show the proportion of variance explained by treatment (RV v. vehicle), status (case v. control), and individual. In the PCA of the RNA-seq data, RV treatment is highly correlated with the first PC, asthma is correlated with PCs 2-3, and sample_ID (individual donor) with PCs 2, 3, 5, 7-10, the latter reflecting individual donor effects. We note that for PC2 (explaining 13% of the variance), case-control (asthma) has a much larger effect than individual. In the PCA of the ATAC-seq data, RV treatment is correlated with PCs 2-5, asthma with PCs 2 and 4, and individual donor with PCs 2, 4, 5, and 10. To account for “individual effects” in our modeling of both ATAC-seq and RNA-seq data, we used the limma option "block" to pair each donors' samples. We then used duplicateCorrelation to account for the replicates. Thus, this correction should adjust the model for inter-individual variation that is not due to RV or asthma status. Blocking is now more clearly described on lines 386 and 416.

PCA of RNA and ATAC-seq analyses. First 10 PCs shown with significant correlations after multiple testing corrections for all covariates indicated in bolded font.

RNA-seq	PC1	PC2	PC3	PC4	PC5	PC6	PC7	PC8	PC9	PC10
% variance explained	18%	13%	10%	7%	5%	5%	3%	3%	2%	2%
RV v Vehicle	3.87E-24	3.19E-03	5.97E-03	7.60E-01	1.67E-01	6.55E-01	8.73E-01	7.76E-01	1.90E-01	5.32E-01
Case v Control	5.57E-01	1.89E-08	5.88E-11	4.28E-01	5.67E-03	3.20E-01	7.49E-01	9.16E-01	2.10E-01	9.41E-02
Sample_ID	9.79E-01	1.81E-04	5.86E-14	2.34E-02	5.01E-17	8.33E-03	6.42E-26	2.09E-18	9.61E-05	1.91E-08

ATAC-seq	PC1	PC2	PC3	PC4	PC5	PC6	PC7	PC8	PC9	PC10
% variance explained	9%	3%	2%	2%	2%	2%	2%	2%	2%	1%
RV v Vehicle	4.52E-02	4.75E-06	1.31E-03	6.57E-05	2.41E-03	1.82E-01	9.65E-01	5.43E-02	6.80E-01	8.71E-01
Case v Control	1.02E-01	6.73E-08	6.45E-01	8.81E-07	2.78E-01	6.83E-01	5.96E-01	2.29E-01	4.15E-02	2.54E-01
Sample_ID	5.24E-03	2.60E-07	3.02E-03	4.54E-04	6.26E-06	3.46E-03	8.90E-01	2.54E-02	5.48E-03	3.64E-15

Associated, the authors should provide statistics for Figure S1.	The FDR adjusted p-values (q) are now provided for Figure S1.
Line 116 on (ATAC-seq analysis) – I have a few comments here. 1. The authors should show some of this ATAC-seq data. What are the quality scores (FRiP or TSS enrichment)? How well do data sets correlate across patient-derived BEC within the control and asthma groups?	Thank you for this comment. Our mean FRiP score across samples is 7.2%, however, we see robust enrichment of ATAC-seq peaks in functional regions; 17.7% of the ATAC-seq peaks are at Epigenome Roadmap predicted transcriptional start sites and 70% of peaks overlap with ENCODE_ChIP-seq predicted transcription factor binding sites. This information has now been added to the methods on lines 409-411, as follows: “The fraction of reads in peaks, or FRiP score, was 7.2% across samples, with 17.7% of peaks at Roadmap predicted transcriptional start sites and 70% of peaks overlapping with ENCODE transcription factor binding sites”. Please see response to comment above ("Second, the authors state...")

2. The authors have used statistics to differentially call regions with altered accessibility. What if they took an RNA-centric approach here? For the genes that are upregulated in control with RV treatment, what happens to their promoters? For the asthma group? How do these regions compare between the two groups at the steady state without RV treatment? These kinds of analyses would allow the authors to address the question posed in line 128 – is there a lost response, or are the promoters in asthma patients “primed”?	Thank you for this excellent suggestion. To address this question, we used the 182 RV responsive gene-differentially responsive ATAC pairs (determined by pcHi-C) in the controls but were not in cases (Supplementary Tables 8 and 9). Among those interacting pairs, we focused on 118 pairs in which the gene was RV-responsive in both cases and controls but the chromatin was only responsive in the controls. Two examples (selected for their asthma relevance) are now shown in Figure 5 (new panels D-E).
3. The authors should provide details of their sequencing experiments – how many reads per sample, data normalization, correlation between data sets, etc. I may have missed this.	These data had been included in the methods. We have now divided this section into subsections. The RNA-seq experiments are described on lines 369-384 and the ATAC-seq experiments on lines 395-417.
4. The figure legend for Figure 4 is incorrect and incomplete. The figure cannot be understood from the figure legend alone.	We apologize for the incomplete figure legend. The legend has been completely rewritten to help with clarity, and now says “Figure 4: HOMER motif analysis shows enrichments of TFBS in areas of RV-responsive open chromatin. A. RV-responsive areas of open chromatin have been separated by those with increased accessibility and decreased accessibility in the RV-treated sample in the first column of the table. The number of enriched TFBS and the corresponding number of unique TF identified by HOMER (in parentheses) are shown in the second column of the table. The last column indicates the number (percent in parentheses) of unique transcription factors identified by HOMER that are also RV-responsive. B. The five most enriched TFBS are shown for areas with increased accessibility in RV treated samples from controls. The black bars indicate the % of regions containing the motif among all ATAC-seq peaks and the grey bar indicates the % of regions containing the motif in the RV-responsive areas of open chromatin. Panels C and D show the five most enriched TFBS for the areas with decreased accessibility in controls and increased accessibility in cases, respectively. *HOMER identified TFBS that correspond to transcription factors that also have RV-responsive gene expression.”
5. Also, here the authors have the opportunity to better integrate their data sets. They make the point that some of the TFs whose binding motifs are enriched from their HOMER analysis are dysregulated, but are they upregulated? Downregulated? How much? Is this a greater trend for the HOMER analysis integrated with RNA-seq, or just a handful of factors? Is the identity of these TF interesting from the perspective of lung function or related viral response?	We agree with the reviewer on this comment. We had this information in an earlier draft and removed it because we thought it might distract from the flow of the paper. We have now added an additional paragraph describing these data (lines 138-153), focusing specifically on the information suggested by the reviewer.

Line 149 - Was there a reason for not using the same lines for the RNA/ATAC and pcHiC?	Ideally, we would have used the same samples for both experiments. However, we initiated the Hi-C studies years before we began the cell culture studies reported in this manuscript. To obtain the number of cells needed for Hi-C at the time those studies were initiated, cells were purified and cryopreserved from donor lungs over a two-year period. We used those ex vivo BEC Hi-C results to capture the diversity in fully differentiated cells. We consider this a minor limitation and now discuss this in the limitations paragraph in the discussion (lines 324-329). “Third, although we use pcHi-C to wire the differentially responsive RNA-seq and ATAC-seq peaks, the pcHi-C was performed in cells from different individuals as those used in the cell culture studies and in ex vivo cells compared to cultured cells in this study. Differences between the donors with respect to sex ratio or other features, or differences in ex vivo vs. in vitro cells may have resulted in our underestimating the number of interactions present in the cultured and RV treated cells.
Line 173 – I’m not a GWAS expert, but these results seem modest. Hopefully, the editors have recruited another reviewer that can better comment on this section of the manuscript.	These results are modest. However, we expected that only a subset of asthma GWAS loci would be attributed to impaired RV-responses given the heterogeneity of asthma and of the large samples required of GWAS. Moreover, we focused only on loci that met stringent criteria for genome-wide significance ($p < 5 \times 10^{-8}$). Therefore, SNPs influencing the expression asthma-associated genes in only a subset of the sample may not reach this level of significance in a GWAS of “asthma”. Overall, therefore, we were not surprised at the modest overlap between RV-responsive genes and asthma GWAS loci.
Reviewer 2	
General 1. There is poor differentiation between types of fold-changes. In every case be clear whether you mean a natural log, log base 2, or log base 10. This is particularly important because the VOOM results in several places are reported as “logFC”, which leaves the reader guessing which is the correct scale.	Thank you for pointing out this omission. We now indicate the log base for each log transformed value throughout the manuscript.
Introduction 2. “... such as asthma, COPD, cystic fibrosis, ...” Abbreviation used without definition. Suggest changing to chronic obstructive pulmonary disease since it doesn’t appear to be used again.	This has been changed (line 33).

3. "... majority of chromatin-chromatin interactions ship at least one gene..." You detected ~600k chromatin – promoter interactions. What percent of your data skip at least one gene? Not strictly necessary, just curious.	11% of the autosomal loops interact with the nearest gene, with a mean of 6 and median of 3 skipped genes per interaction. Our results are consistent with reports in other cells types (e.g., Montefiori, L. E. et al. A promoter interaction map for cardiovascular disease genetics. Elife 7, doi:10.7554/eLife.35788; 2018). Thank you for your interest in these data.
4. "... revealed 193 regions... that were correlated with the expression of RV-responsive target genes..." How did you define this? Maximum distance between gene start / end and ATAC peak? Overlap between ATAC and gene start end? Must be the closest gene to the ATAC peak?	The RNA-ATAC pairs were defined by pcHi-C looping. This was described starting on line 166 "To further characterize these regions, we defined the pcHi-C-predicted target genes for RV-responsive areas of open chromatin, and then tested for correlations between the relative 'openness' of the chromatin and the transcript levels of the predicted target gene.
5. "... we performed 100 permutations in which 4,645 or 3,381 genes were randomly selected from..." How? Python? R? Another tool?	We used R for the permutations, which we added to the sentence on line 210 "To test whether the overlap of RV-responsive genes and GWAS loci were more than expected by chance, we performed 100 permutations in R in which 4,645 or 3,381 genes were randomly selected from the 13,629 genes detected as expressed in BECs"
Methods 6. "RNA- and ATAC-Sequencing, QC and analysis" Suggest splitting this into a sub-section on RNA-Seq and a sub-section on ATAC-Seq. Less confusing that way.	We agree, this has now been separated into two sections.
7. "...completed using the Illumina Nextera XT DNA Library Preparation Kit..." Obviously a DNA kit doesn't work for RNA. It would for cDNA. Did you use a kit or homemade method for cDNA generation? Was it ribosomal depletion or poly(A) mediated? Or did you use a different kit for RNA-Seq?	We have corrected this omission on lines 370-371 by stating "RNA was converted to cDNA using the SMART-Seq v4 Ultra Low Input RNA kit for sequencing (Takara Bio cat. 634898)".
8. "...reference sequence using STAR." Version of RNA-STAR missing.	The version of STAR (2.6.1) is now included in the methods (line 377)
9. "PCA was used to select covariates..." How? R? If so prcomp or princomp? Which R version? Same issue in the ATAC-Seq section.	We used prcomp in R (version 3.6.2). This has been clarified for both the RNA and ATAC-seq methods sections on line 378 and line 411.
10. "... the transposition reaction was incubated..." Commercial TN5, kit version, AddGene, or other?	This information has been added to the following sentence in the methods section (lines 391-392) "For each sample, 5,000 cells were processed immediately using the Nextera DNA Library Prep Kit (Illumina FC-121-1030) ...".
11. "...Qiagen MinElute PCR Purification kit was use to clean up transposition reaction." No bead size selection?	We did not do bead selection.
12. "ATAC-seq peak calls were determined using MACS2." No version.	The version of MACS2 was 2.1.0 and has been added to the methods (line 403).

13. “Peak calls from all samples were pooled across all samples and overlapping...” How? MACS2? Bedtools? R? Python? Version?	We used bedtools version 2.26.0, this has now been clarified in the methods (line 403).
14. “Advaita iPathwayGuide was used to determine...” No release / version number.	The version v1906 has been added to line 422.
15. “Significance was defined as a CHiCAGO score > 5.” Is this their native negative log-weighted p-value, i.e. significance was P approximately 10^{-5}?	That is correct, we have now included a reference (line 444).
Data availability 16. “... will be deposited in GEO.” Publication should obviously not happen before the data are deposited in GEO or dbGAP. Sequencing data is identifiable and depending on the approvals surrounding these donations GEO might not allow their deposition.	All ATAC-seq, RNA-seq and pcHi-C data are already in GEO and the accession number for the study folder has been added to the methods (lines 474-476).
Figures 17. Figure 1 – The x-axis is just log fold change. What log scale? The y-axis is $-\log_2$ p-value. This is a non-standard volcano plot scale, which usually uses $-\log_{10}$ adjusted P. I believe this is a typo. $-\log_2(0.05)$ is 4.32. $-\log_{10}(0.05)$ is 1.30. The horizontal line sure *looks* closer to 1 than 4. “The absolute log fold change...” there are negative values, so it can’t be absolute fold-change.	You are correct, we have now clarified log base values throughout the paper. Thank you for your careful read.
18. Figure 2 – Again shows a $-\log_2$(adj. P) scale. Is it log2 or log10? In sub-panel D, I understand the consistent color usage to help tie figures together. Personally I find the alternating colors hard to read in this plot. Purely personal preference, but you should consider not highlighting or just using an alternating light gray.	The log base has been added and the color for sub-panel D in Figure 2 has been changed to grey to make it easier to read.
19. Figure 3 – Another unspecified log fold-change.	This has been corrected.
20. Figure 4 – This is overall confusing. I try to reach the metric that each figure should be interpretable without the accompanying text in the main document. The legend of 4 doesn’t help much. The sub-panel letters don’t match the figure. You need to tell the reader what the different bars mean (black is genomic background frequency?) in B-D. In table 4a there are some spacing issues in the “enriched TFBS (unique TF)” column where there is no space between the number and the parenthetical. You might want to change the last sentence to something like “Asterisks indicate transcription factors that have significant differential expression between cases and controls...”	We apologize for this oversight. To have now fixed the spacing issues and simplified 4A by removing data that were not discussed in the paper (and should have been removed from previous version). We have rewritten the figure legend and hope this is clearer to the reviewers. Also see response to Reviewer 1 comment 4 above. “Figure 4: HOMER motif analysis show enrichment of TFBS in areas of RV-responsive open chromatin. A. RV-responsive areas of open chromatin have been separated by those with increased accessibility and decreased accessibility in the RV-treated sample, the first column of the table, prior to running HOMER motif analysis. The number of enriched TFBS and the

	corresponding number of unique TF identified by HOMER (in parentheses) are noted in the second column of the table. The last column indicates the number (percent in parentheses) of unique transcription factors identified by HOMER that are also differentially expressed in the RV treated samples. B. The top five most enriched TFBS are shown for the areas with increased accessibility with RV treatment in controls. The black bars indicate the % of regions containing the motif across the background (all ATAC-seq peaks) and the grey bar indicates the % of regions containing the motif in the RV-responsive areas of open chromatin. Panels C and D show the top five most enriched TFBS for the areas with decreased accessibility in controls and increased accessibility in cases, respectively. *HOMER identified TFBS that correspond to transcription factors that also have RV-responsive gene expression.”
Introduction 21. “... in the presence of RV-A16 or vehicle...” Suggest something like “... in the presence of rhinovirus strain RV-A16 or vehicle...” for clarity.	Thank you. We have now made this change on line 51.
23. Many of the supplemental table fields I only understand because of familiarity with these types of analysis. Suggest adding a supplemental document that for each table defines each field to remove any ambiguity for non-experts.	We have added an additional supplemental table (#12) describing the columns for the supplemental tables requiring more description.
24. May be easiest to combine *all* supplementary tables into a single workbook with multiple sheets, rather than dividing among CSV, workbook, and PDF.	We organized the tables this way according to the journal’s requirement for each supplemental file to be submitted separately (and as PDFs if they fit). We would be happy to combine the excel tables into one workbook with multiple sheets if that is permissible by the journal.
25. Supplementary Table 3 – as an aside, we similarly sometimes see discordant fold-changes. I agree that for the sake of transparency it’s worth listing them. Quick examination of the p-values show adjusted Ps quite close to the cutoff. Probably false-positives.	Thank you for this comment. We agree with the reviewer that the genes responding to RV in opposite directions in the cases and controls are only modestly significant and likely to be false positives.
26. Supplementary Table 4 – Suggest reducing the table to only ontologies significant in one or the other. I tend to keep full gene lists for differentially expressed genes. But the large number of p-value == 1.0 make it difficult to sift through this table.	This table is now limited to GO terms that are significantly enriched in either cases or controls (or associated with asthma in the vehicle treated or RV treated as requested).
27. Supplementary Tables 9 and 10 – might be helpful to include the annotated gene start and end so we know how close these peaks are (or if they’re overlapping with) the gene.	We agree. Chromosomal locations for the genes have now been added to these tables.
Reviewer 3	

This study is conducted upon basal cells isolated from subject lungs, could the authors describe whether there were compositional differences between the cells obtained from either individual donors or between subject groups?	Cell composition was not assayed at the time of collection. However, basal cells are the only type of airway epithelial cell that replicates in culture (Everman, 2018, Methods Mol Biol). All cells were passage 2-3, therefore any remaining cell types should be in negligible proportions (if at all) in our model.
Cells from different donors were used for the HiC experiments, these donors had a different ratio of gender and ethnicity. Could the authors please comment in the implications of this for data interpretation.	We added to the limitation paragraph in the discussion (324-329) that we used different cells from different donors for the Hi-C studies. “Third, although we use pcHi-C to wire RV-responsive ATAC-seq peaks to their RV-responsive gene targets, the pcHi-C was performed in cells from different individuals as those used in the cell culture studies and in ex vivo cells compared to cultured cells in this study. Differences between the donors with respect to sex ratio or other features, or differences in ex vivo vs. in vitro cells may have resulted in our underestimating the number of interactions present in the cultured and RV treated cells”
There was quite an age range in the donors, did the authors observe any epigenetic differences between cells from young and old and how might any differences these contribute to interpretation of data	Age is not significantly correlated with any of the top 10 PCs for the ATAC data ($p = 0.45, 0.01, 0.27, 0.02, 0.23, 0.09, 0.65, 0.03, 0.07, 0.62$; multiple testing correction suggested p value threshold = 0.0025), suggesting that in our data, age is not driving any global trends. Based on the PCA, we did not adjust for age in our model.
could the authors describe any medication that the donors were on prior to death.	Data on medication use in the lung donors come from the Gift of Hope Regional Organ Bank of Illinois and are not always complete. However, we have now included the data that was provided to us in Supplementary Table 1.
Throughout the manuscript it is not clear what inherent differences there are between the study populations prior to rhinoviral infection. If there are fundamental and consistent differences in epigenetic landscapes between the subject groups then it follows that transcriptomes and ATAC profiles will be different following infection. Thus, could the authors describe which pathways were differentially expressed at baseline in order to separate asthma from infection.	Thank you for this comment, we also thought this was an important comparison. These data (comparison of asthma cases vs controls with and without RV infection) were included in the supplement for both the RNA-seq and ATAC-seq data. We have now added pathway analysis to Supplementary Table 4 and refer to this table on line 423-424.
In figure 4 data shows enriched TF motifs in ATAC regions, in response to RV. Were any of these motifs found in combination? What percentage of regions lacked any of the motifs shown ?	Given that the percent of regions with different motifs add up to more than 100% across the different tested motifs, we assume that there is overlap of some motifs. Unfortunately, this information can't be extrapolated from the Homer motif analysis in which the input regions are first parsed into smaller regions and then fed into the analysis. Therefore, the output is based on percent of these smaller regions, not ATAC-seq peaks, and we cannot determine the percent of peaks lacking motifs. However, a comparison of the ATAC-seq peaks library to known ENCODE transcription factor binding sites revealed that 70% of all peaks contain transcription factor binding sites (line 410).

Figure 5

Figure 5: RV-responsive areas of open chromatin correlated with gene expression at pHi-C predicted target genes. **A.** The numbers of RV-responsive genes, chromatin regions, or pHi-C target genes of RV-responsive chromatin regions in controls (gray) and cases (burgundy) and both (white) are shown in the first three columns. The last column shows the number the pHi-C predicted genes where the chromatin accessible and gene expression were correlated (Spearman’s correlation, FDR adj. $p < 0.05$). **B.** The promoter of *PLAUR* loops to an area of open chromatin at chr19:43967179-43968550. Chromatin accessibility for each condition is shown in the top panel, gene expression in the second panel and the bottom panel indicates pHi-C looping from the *PLAUR* promoter. The pink box indicates the *PLAUR* promoter, the blue box indicates the RV-responsive area of open chromatin. **C.** Correlation between expression of *PLAUR* and chromatin accessibility at chr19:43967179-43968550. **D-E** and **F-G** are gene-chromatin pairs where the gene is RV-responsive in both cases and controls while the chromatin is only RV-responsive in the controls.

Reviewers' Comments:

Reviewer #1:

Remarks to the Author:

Generally, the authors have addressed my concerns. One additional point is that in the new analysis in Figure 5, it would be nice to see whether the behavior holds across the 189 pairs identified from the control group. For example, split the pairs into regions that become more accessible or less accessible in RV treatment and then ask for those two groups what happens with the asthma patients. The examples provided are striking, but it's important to ask about the prevalence.

Reviewer #2:

Remarks to the Author:

I appreciate the authors taking the time to address the multiple reviewer comments. I believe the manuscript is now more readable and more accessible than it was in the first iteration. All of my comments have been satisfactorily addressed.

Reviewer #3:

None

We would like to thank the reviewers again for their careful and thoughtful read of our manuscript as well as in their interest in the figure added after the last revision.

Reviewer Comment	Response
Generally, the authors have addressed my concerns. One additional point is that in the new analysis in Figure 5, it would be nice to see whether the behavior holds across the 189 pairs identified from the control group. For example, split the pairs into regions that become more accessible or less accessible in RV treatment and then ask for those two groups what happens with the asthma patients. The examples provided are striking, but it's important to ask about the prevalence.	We have now addressed this comment in detail on lines 187-199. Of the 189 gene-chromatin pairs, 119 were not correlated in cases because the chromatin is not responsive, even though the corresponding gene is responsive. Remarkably, 45% of the chromatin accessibility sites from these 119 pairs (54 pairs) were not even nominally (unadjusted $p < 0.05$) responsive to RV, suggesting that the chromatin is already in a primed orientation. For the remaining 65 areas of open chromatin, the nominal $p < 0.05$ suggests a trend towards an RV-response, although it's possible that some of those regions are also primed.